A comprehensive review of oncogenic Notch signaling in multiple myeloma

Roosma Justin jroosma@ewu.edu
Biology, Eastern Washington University , Cheney, Washington , United States
Uversky Vladimir
Electronic publication date: 2024 Nov 28
Publication date: 2024
Volume: 12
Electronic Location ID: e18485
Received 2024 May 1; Accepted 2024 Oct 16
Copyright: © 2024 Roosma
Copyright year: 2024
Copyright holder: Roosma
License: This is an open access article distributed under the terms of the Creative Commons Attribution License, which permits unrestricted use, distribution, reproduction and adaptation in any medium and for any purpose provided that it is properly attributed. For attribution, the original author(s), title, publication source (PeerJ) and either DOI or URL of the article must be cited.
License URL: https://creativecommons.org/licenses/by/4.0/

Keywords: Multiple myeloma, Notch signaling, Tumor microenvironment, Plasma cells, Signal transduction, Bone marrow metastasis, Leukemia, Cancer genetics, Precision medicine, Cancer biology

Funding: The authors received no funding for this work.

==============================
Multiple myeloma remains an incurable plasma cell cancer with radical case-by-case heterogeneity. Because of this, personalized and disease-specific biology of multiple myeloma must be understood for the discovery of effective molecular targets. The highly evolutionarily conserved Notch signaling pathway has been extensively described as a multifaceted driver of the multiple myeloma disease process—contributing to both intrinsic effects of malignant cells and to widespread remodeling of the tumor microenvironment that further facilitates disease progression. Namely, Notch signaling amongst malignant cells promotes increased proliferation, tumor-initiating capacity, drug resistance, and invasiveness. Moreover, Notch signaling between malignant cells and cells of the tumor microenvironment leads to increased osteodegenerative disease and angiogenesis. This comprehensive review will discuss both the intrinsic implications of pathological Notch signaling in multiple myeloma and the extrinsic implications of Notch signaling in the multiple myeloma tumor microenvironment. Additionally, the genetic origins of Notch signaling dysregulation in multiple myeloma and current attempts at targeting Notch therapeutically will be reviewed. While the subject has been reviewed previously, recent developments in the intervening years demand a revised synthesis of the literature. The aim of this work is to introduce and thoroughly synthesize the current state of knowledge in this vein of research and to highlight future directions for both new and in-the-field scientists.

Survey Methodology

A comprehensive and unbiased search was conducted to identify all published biological literature documenting the implications of Notch signaling in multiple myeloma since Notch was first characterized as an oncogenic signaling pathway in the disease 20 years ago. Criteria for inclusion required discussion of multiple myeloma or Notch signaling or biological/therapeutic mechanisms relevant to the discussed subject matter. Only academic literature was cited with primary literature and dissertation writing taking precedence over review articles. Search terms for the compiled literature included but were not limited to, “multiple myeloma, oncogenesis, Notch signaling, B cell development, malignant proliferation, bone marrow metastasis, extracellular vesicles, tumor microenvironment, bone disease, angiogenesis, drug resistance, cancer immunology, and Notch targeting therapeutics”. Literature was queried and accessed on PubMed, Scopus, Google Scholar, and Google.

Introduction

Multiple myeloma (MM) is a blood cancer of plasma cells, which are terminally differentiated antibody-secreting B lymphocytes. MM is the second most common hematologic malignancy behind Non-Hodgkin’s lymphoma (Jemal et al., 2010). In 2018, there were an estimated 160,000 cases and 106,000 mortalities associated with MM globally. In 2024, it was estimated that there would be 35,780 new diagnoses and 19,520 US deaths associated with MM (Siegel, Giaquinto & Jemal, 2024). Global MM incidence has risen by 126% since 1990 but the disease predominates in Western countries, with New Zealand and Australia having the highest incidence followed by the US and various European nations (Padala et al., 2021; Ludwig et al., 2020; Cowan et al., 2018). The disease disproportionately affects men, as they are 1.5 times more likely to acquire the disease than women; the elderly, where the average age of diagnosis is 69; and African Americans, who are >2 times more prone to developing the disease in the US (Padala et al., 2021; Waxman et al., 2010). Acquisition of MM is associated with heritable genetics, obesity, chronic antigen stimulation, and exposure to ionizing radiation or environmental carcinogens in agricultural occupations (Kundu et al., 2022).

MM tumors primarily manifest as intramedullary disease in the bone marrow, but in some cases, it may also affect secondary lymphatic organs as extramedullary disease (Wiedmeier-Nutor & Bergsagel, 2022). Symptomology for MM is often described by the acronym SLiM-CRAB: S, referring to a population of 60% clonal plasma cells in the bone marrow; Li to serum free light chain levels of >100 mg/L; M to at least 1 MRI detected bone lesion; C, referring to hypercalcemia due to the increased degradation of mineralized bone; R to renal disease caused by hypercalcemia and over-secretion of monoclonal immunoglobulins; A to anemia as proliferating MM cells crowd the normal process of bone marrow erythropoiesis; and B to bone disease as MM cells perturb the normal balance of bone remodeling in favor of degradation which leads to lesions, bone pain, and fragility fractures in the majority of patients (Rajkumar, 2016).

Staging of MM begins with monoclonal gammopathy of undetermined significance (MGUS) which is characterized by elevated clonal immunoglobulin levels but with bone marrow clonal plasma cell (BMPC) levels of <10% and the absence of CRAB symptoms. Smoldering multiple myeloma (SM) is characterized by >10% BMPC but no significant CRAB symptoms. Multiple myeloma is officially diagnosed when BMPC levels are >10%, and multiple SLiM-CRAB criteria are met (Padala et al., 2021; Rajkumar, 2024). Finally, plasma cell leukemia (PCL) is an advanced stage of MM that is defined by >5% of circulating clonal plasma cells (Rajkumar, 2024; Gundesen et al., 2019). 5-year survival rates of SM are 79% and for fully diagnosed MM, 57% (Padala et al., 2021; American Cancer Society, 2023). For a more extensive review on the clinical criteria of the international working group diagnostic criteria, see work done by Rajkumar (2024).

Primary and secondary molecular events of MM

Immense genetic variation is observed between MM cases, as there is no single event that leads to the disease. However, primary events leading to the onset of MM are broadly classified as hyperdiploidy (HD) and/or non-hyperdiploidy.

HD occurs in 50% of MM cases and is a phenomenon in which greater than two chromosomal copies exist. In MM, HD most often presents as trisomies of odd-numbered chromosomes (3, 5, 7, 11, 15, 19, and 21) (Wiedmeier-Nutor & Bergsagel, 2022; Colombo et al., 2015). It is poorly understood what causes HD and how HD drives disease progression in MM. It is speculated that HD may occur during germinal center formation and that HD may lead to the onset of MM by dysregulation of genes found in excess copy number. However, HD inevitably leads to the dysregulation of thousands of genes making it difficult to pinpoint the exact contribution to MM oncogenesis (Wiedmeier-Nutor & Bergsagel, 2022).

Non-HD MM involves aberrant chromosomal translocations, most often at the immunoglobulin heavy chain (IgH) locus of chromosome 14. These translocations position oncogenes as downstream targets of powerful IgH enhancers. The most common of which are t(11;14), t(4;14), t(6;14), t(14;16), and t(14;20), leading to the primary dysregulation of cyclin D1, fibroblast growth factor 3 (FGFR3)/multiple myeloma SET domain (MMSET), cyclin D3, C-MAF, and MAFB, respectively. Exclusive non-HD MM accounts for 60% of MM cases but 90% of all cases experience translocation events. Non-HD MM is generally associated with worse outcomes as compared to HD MM and the number of translocations generally correlates with the severity of non-HD MM (Wiedmeier-Nutor & Bergsagel, 2022).

Translocations leading to MM result from the inherent chromosomal instability of plasma cell biology. B cells undergo variable diverse joining recombination (VDJ recombination) of chromosome 14 at the variable, diversity, and joining segments of the IgH locus to create antibody binding domain diversity in pro-B cells (Christie, Fijen & Rothenberg, 2022). Then, mature naïve B cells, migrate to secondary lymphoid tissue where in response to their complementary antigen, they are activated into germinal center plasma cells. These differentiating pre-plasma cells undergo somatic hypermutation—the process of fine-tuning antibody specificity through random point mutations, and class switching recombination (CSR)—the process of changing the class of expressed antibody (Roth et al., 2014; González et al., 2007). CSR requires double-stranded breaks and linkages at the IgH locus which creates susceptibility to the translocation events that give rise to MM (González et al., 2007).

In addition to primary events, many secondary events contribute to further disease progression which may manifest as chromosomal duplications, deletions, single nucleotide variants (SNVs), and epigenetic changes. Of particular significance are chromosome 17 deletion, gain of the q arm of chromosome 1, RAS mutations, and MYC structural variants (Wiedmeier-Nutor & Bergsagel, 2022). Chromosome 17 deletion involves deletion of the core tumor suppressor gene p53 and gain of the q arm of chromosome 1 leads to the increased copy number of the oncogenes MCL1 and CKS1B (Wiedmeier-Nutor & Bergsagel, 2022; Burroughs Garcìa et al., 2021).

The Notch signaling pathway

The Notch signaling pathway becomes dysregulated as a secondary event in MM. As a consequence, Notch acts as a multifaceted driver of tumor growth and other disease aspects by broadly reprogramming gene expression in malignant and non-malignant cells of the MM tumor microenvironment (Colombo et al., 2015).

Notch is highly conserved as it is found in all metazoan species; it serves pleiotropic roles in cell and tissue development contributing to cell determination, proliferation, or apoptosis in a context-dependent manner (Kopan & Ilagan, 2009). In human embryogenesis, Notch signaling is known to be involved in neural, somite, and vascular development. In adults, it is primarily involved in tissue renewal in various organ systems such as the intestines, skin, blood, liver, kidneys, nervous system, bone, and muscles (Sato et al., 2012).

In mammals, there exist four variants of the Notch receptor consisting of Notch1, 2, 3, and 4; two families of ligands, the serrate family consisting of Jagged1 and Jagged2; and the Delta-like ligand family consisting of Delta-like ligand 1, Delta-like ligand 3 and Delta-like ligand 4 (DLL1, 3, 4). While variation exists between the Notch receptors and ligands, all share characteristic features (Kopan & Ilagan, 2009).

Before membrane integration, Notch receptors are post-translationally modified by: O-Fut1, Rumi, and the family of Fringe enzymes to modulate receptor-ligand sensitivity and specificity (Kopan & Ilagan, 2009; Kakuda et al., 2020). The receptor is then cleaved at site 1 (S1) by a furin-like convertase to create a noncovalent association between the extracellular and intracellular regions (Fig. 1) (Kopan & Ilagan, 2009).

Figure 1 The Notch signaling pathway.

Notch cell-to-cell interaction between a sending (purple membrane) and receiving cell (blue membrane). Notch receptor components: EGF (epidermal growth factor repeats), NRR (negative regulatory region), HD (heterodimerization domain), RAM (RBPjκ associated motif), ANK (Ankyrin domain), TAD (transactivation domain), and PEST (proline, glutamate, serine, threonine rich domain). Notch ligand components: EGF (epidermal growth factor repeats), and DSL (delta serrate ligand). Adapted from Biorender.com.

The extracellular domain of the Notch receptors is involved in ligand binding and is composed of epidermal growth factor-like (EGF) repeats, the negative regulatory region (NRR), the heterodimerization domain, and the hydrophobic transmembrane domain (Kopan & Ilagan, 2009). The intracellular domain of the Notch receptor is composed of the RBPjk association molecule (RAM), the ankyrin repeat domain (ANK) flanked by two nuclear localization sequences, and the transactivation domain (TAD) which collectively facilitate binding to the transcription complex upon intracellular release. Finally, the PEST sequence of the intracellular domain acts as a marker for cytosolic degradation (Fig. 1) (Kopan & Ilagan, 2009; Garis & Garrett-Sinha, 2021).

The extracellular domains of the Notch ligands consist of the Delta-Serrate-Lag-2 (DSL) region, the EGF region responsible for receptor interactions, and the transmembrane region of the ligand. For most Notch ligands, the intracellular domain of the Notch ligands is composed of a lysine-rich region and a PDZ ligand motif which facilitates adhesion to cytoskeletal components (Fig. 1) (Platonova et al., 2016).

Notch receptor activation is initiated by contact-dependent interactions between cells. The ubiquitin ligases Mindbomb and Neuralase then mark the intracellular domain of the Notch ligand for degradation and subsequent endocytosis by the proteasome—a process necessary to induce conformational activation of the Notch receptor (Kopan & Ilagan, 2009). The extracellular S2 site of the receptor is subsequently exposed and cleaved by ADAM metalloprotease leaving the Notch extracellular truncation fragment (NEXT) to be cleaved by gamma-secretase at the transmembrane sites 3 and 4 (S3/S4). The gamma-secretase complex is composed of four domains: nicastrin (NCSTN), anterior pharynx defective 1 (APH1), presenilin enhancer 2 (PSENEN), and the catalytic domain presenilin (PSEN1/2) (Serneels et al., 2023).

After cleavage, the Notch intracellular domain (NICD) localizes to the nucleus where it associates with the DNA binding complex CBF1/Su(H)/Lag01 (CSL) thereby disinhibiting it from transcriptional repressors histone deacetylase (HDAC) and corepressor R (Co-R). The transcription activator mastermind like (MAML) and other coactivators (Co-A) are then recruited allowing for transcription of the target family genes HES (Hairy and enhancer of split) and HEY (Hairy and enhancer of split related to the YRPW motif) (Fig. 1) (Kopan & Ilagan, 2009).

Notch dysregulation

While the genetic origins of Notch dysregulation in MM are not completely understood and may vary between cases, its involvement in MM is widespread as 92% of patient samples express Notch1 and 92% of patient samples also express Jagged1 whereas normal post-germinal center plasma cells lack expression of these genes (Garis & Garrett-Sinha, 2021; Škrtić et al., 2010).

In terms of major molecular events, HD MM may lead to dysregulated Notch signaling due to increased chromosomal copy number leading to overexpression of Notch1 (chr9), Notch3 (chr19), DLL3 (chr19), DLL4 (chr15), MAML1 (chr5), and MAML2 (chr11) (Colombo et al., 2015). For non-HD MM, overexpression of Notch can be attributed to translocations t(14;16) and t(14;20) which lead to the dysregulation of the transcription factors C-MAF and MAFB which both lead to downstream overexpression of Notch2 (Colombo et al., 2015; van Stralen et al., 2009). It has also been discovered that chromosomal deletion 17p13, and consequent deletion of p53 is associated with an upregulation of Notch signaling components, Notch1, Jagged1, and HES1 (Chang et al., 2023). Notch1 overexpression in MM may also be attributed to loss of function of the tumor suppressor micro-RNA, miR-125b. Mechanistically, miR-125b targets the 3′ end of the long noncoding RNA (lncRNA) MALAT1 (metastasis-associated lung adenocarcinoma transcript) which is a Notch 1 stabilizer. Loss of function of miR-125b leads to increased MALAT1 mediated stabilization of Notch1 (Gao et al., 2018).

Overexpression of the Notch ligands have been demonstrated as clinical biomarkers of disease severity in MM. The transition from MGUS to MM is marked by the overexpression of Jagged1 and Notch1 (Škrtić et al., 2010). Jagged2 overexpression has been demonstrated as a biomarker for the onset of MGUS (Houde et al., 2004). In vivo and ex vivo evidence have all demonstrated Jagged1/2 correlation with MM tumor burden (Jundt et al., 2004; Platonova et al., 2023). Jagged1 overexpression has been reported to be driven by downregulation of the 3′ UTR-targeting tumor suppressor micro RNAs, miR-186 and miR-26b-5p. Reduced expression of these micro-RNAs is correlated with worse clinical outcomes and increased malignant cell proliferation (Liu et al., 2016; Dong & Zheng, 2024; Jia et al., 2018). Jagged2 overexpression may occur due to hypomethylation of the Jagged2 promoter, the absence of the SMRT/NCoR2 corepressors from the Jagged2 promoter, and become further potentiated by the overexpression of the ubiquitin ligase skeletrophin which is involved in catalyzing Jagged2 signaling (Colombo et al., 2015; Houde et al., 2004). Jagged2 then leads to further disease progression by increasing MM cell proliferation and its expression is correlated with disease severity (Houde et al., 2004; Platonova et al., 2023; Chiron et al., 2012).

Notch in B cell development

B cells exhibit variable expression of Notch receptors across their development (Fig. 2). Notch1 appears to be expressed across all stages whereas Notch2 is expressed at the pre and transitional B-cell stages. However, expression of both Notch1 and Notch2 is lost in germinal center B cells. Expression of Notch3 and Notch4 in B cells has not been thoroughly investigated (Garis & Garrett-Sinha, 2021).

Figure 2 Oncogenesis of multiple myeloma in B-cell development.

Plasma cells originate from hematopoietic stem cells (HSCs) in the bone marrow and eventually become pro-B cells, pre-B cells, and then mature naïve B cells before disseminating into secondary lymphoid tissue. There, mature naïve B cells become transitional B cells then marginal zone B cells (MZB) or follicular B cells. Upon antigen presentation, MZB or follicular B cells proliferate and form germinal centers where a fraction of these proliferating germinal center cells return to the bone marrow. During germinal center formation, somatic hypermutation (SHM) and class-switching recombination occur (CSR). Aberrant primary events leading to multiple myeloma, such as translocations, occur during CSR at the switch region of the IgH locus. Further secondary events lead to increased disease severity. Adapted from Biorender.com.

Only one account of B cells expressing Notch ligands has been documented whereas other studies have reported their absence (Garis & Garrett-Sinha, 2021; Zhu et al., 2017). Because of this, it is suspected that the canonical source of ligands leading to Notch activation in B cells primarily involves the splenic and lymphoid stroma. Mechanistically, it has been demonstrated in mice that DLL1 knockdown in radiation-resistant stromal cells and fibroblasts of the spleen impairs MZB cell differentiation (Garis & Garrett-Sinha, 2021; Fasnacht et al., 2014; Sheng et al., 2008). In mouse models, it has been further demonstrated that MZB cells require activation of Notch2 as its knockout prevents their formation (Saito et al., 2003). On the contrary, differentiation of follicular B cells appears to occur independently from canonical Notch signaling as knockout of CSL does not affect their formation (Tanigaki et al., 2002).

Notch plays an important role in the formation of antibody-secreting plasma cells and germinal centers. Activation with DLL1 or NICD1 overexpression has been shown to increase differentiation, class-switching recombination, and in some cases, increased proliferation of plasma cells in a MAML-dependent manner (Santos et al., 2007; Thomas et al., 2007; Kang, Kim & Park, 2014). Moreover, inhibition of, Notch1 and gamma-secretase have been shown to prevent differentiation of MZB and follicular B cells into plasma cells, respectively (Kang, Kim & Park, 2014; Yoon et al., 2009). Remarkably, differentiation of plasma cells has been reported to occur independently of the CSL transcription factor indicating that a non-canonical variation of the Notch pathway governs this transformation (Tanigaki et al., 2002).

Malignant proliferation

Notch signaling dysregulation increases bulk proliferation of malignant MM cells and their tumor-initiating capacity. It has been observed that pan-inhibition of Notch signaling yields reduced MM proliferation and increased cell cycle arrest in vitro (Mirandola et al., 2013). Moreover, MM cell expression of Jagged ligands is correlated with disease stage and stimulation of MM cells with exogenous Jagged ligands has been shown to drive proliferation (Jundt et al., 2004; Platonova et al., 2023).

Homotypic, or autonomous, Jagged1-Notch1 interactions between MM cells have been shown to increase cell proliferation through the upregulation of the downstream effectors cyclin D1 and D2 as blockade of Notch1 with neutralizing antibodies has been shown to reduce their expression in MM cells (Jundt et al., 2004; Ishibashi et al., 2020). Additionally, knockdown of Jagged1/2 in MM has been shown to downregulate autocrine and BMSC expression of interleukin-6 (IL-6)—a canonical growth factor for plasma cells that is normally secreted by CD4+ cells and BMSCs (Colombo et al., 2016; Harmer, Falank & Reagan, 2019). Overexpression of IL-6 is known to correlate with disease severity and tumor growth in MM (Harmer, Falank & Reagan, 2019). Notch-mediated autocrine secretion of IL-6 in MM cells allows for increased independence from the tumor microenvironment (TME) for MM cell survival.

In vitro and ex vivo inhibition of Notch signaling has been shown to significantly reduce tumor-initiating capacity. Gamma-secretase inhibition and HES1 knockdown have both been demonstrated to reduce autonomous colony-forming ability in MM cells enriched in Notch signaling (Chang et al., 2023). Furthermore, in vitro and in vivo xenograft work has demonstrated that Jagged2 correlates with MM colony-forming ability and that disruption of the Jagged2/Notch axes through chimeric molecules antagonizes this (Chiron et al., 2012). Finally, DLL1/Notch interactions have also been shown to increase colony forming and tumor proliferation for in vitro and in vivo models (Xu et al., 2012). Exposure of MM cells to DLL1 overexpressing BMSCs has been shown to stimulate gamma secretase-dependent colony-forming ability and decrease time to mortality for mouse engraftments. Mechanistically, this was suggested to be caused by Notch-mediated reduction of the cycle checkpoint inhibitors p21 and p27 (Xu et al., 2012).

Bone marrow chemotaxis

Like normal plasma cells, MM cells eventually migrate to the bone marrow which serves the malignant growth of MM cells by providing protective niches against drugs and the immune system (Fig. 2) (Garis & Garrett-Sinha, 2021; Ullah, 2019; Coniglio, 2018). The CXCR4/SDF1 (CXC receptor 4/stromal derived factor 1) axis is a ubiquitous chemokine/seven pass GPCR signal transduction pathway represented in MM and many bone marrow metastatic cancers (Ullah, 2019; Otsuka & Bebb, 2008). Plasma cells are normally recruited to the bone marrow for long-term residence through the CXCR4/SDF1 axis, however, MM cells overexpress CXCR4 leading to excessive bone marrow invasion (Mirandola et al., 2013; Ullah, 2019; Zannettino et al., 2005).

CXCR4 activation in MM cells activates pro-invasive genes, such as integrins, that allow for endothelial and bone marrow adhesion; cytoskeletal actin, which must be rearranged to allow for cellular flexibility; and matrix metalloproteinases (MMPs), which promote ECM degradation (Ullah, 2019; Parmo-Cabañas et al., 2004; Alsayed et al., 2007). In MM, expression of Notch1 is correlated with bone marrow dissemination and Notch inhibition has been demonstrated to reduce this (Škrtić et al., 2010; Mirandola et al., 2013). Further investigation has revealed that Notch signaling drives expression of CXCR4 (Mirandola et al., 2013). Moreover, it has been demonstrated that Notch also drives expression of SDF1 in MM, as multiple in vitro studies have demonstrated that both Notch pan inhibition and knockdown of Notch2 reduce its expression (Mirandola et al., 2013; Maichl et al., 2023). Finally, knockdown of MM Jagged1 and 2 reduces paracrine expression of SDF1 in BMSCs. Together, these findings may suggest the occurrence of Notch-mediated positive feedback between colonized MM cells that recruit circulating MM cells (Mirandola et al., 2013; Maichl et al., 2023).

In addition to metastasis, CXCR4/SDF1 has been demonstrated to contribute to tumor proliferation. In vitro G2/M growth arrest due to pan-inhibition of Notch has been demonstrated to be rescued by exposure to SDF1, thereby providing evidence that Notch-driven tumor proliferation may be partially mediated by CXCR4/SDF1 signaling (Mirandola et al., 2013).

Notch and multiple myeloma extracellular vesicles

Extracellular vesicles (EVs) collectively refer to membrane-bound particles that allow for the stable transport and delivery of various macromolecules to distant cells which may refer to exosomes, microvesicles, or apoptotic bodies (Théry et al., 2018; Saltarella et al., 2021). In MM, EVs have been shown to prepare the bone marrow premetastatic niche for invasion by promoting extravasation, angiogenesis, osteolytic disease, BMSC re-education, and immunosuppression (Saltarella et al., 2021; Colombo et al., 2019; Giannandrea et al., 2022).

Notch1, Notch2, and NICD2 have been demonstrated as active components of the MM-EV cargo. In vitro and ex vivo knockdown of Notch2 has been shown to reduce its presence in MM-EVs and reduce the osteoclastogenic and angiogenic effects of MM-EVs. Additionally, it has been demonstrated, in vivo using a zebrafish xenograft model, that MM-EVs disseminate and upregulate Notch target genes in distant locations (Giannandrea et al., 2022). The micro-RNA, miR-146a, has also been shown to be an active component of MM-EV cargo. In MM, delivery of miR-146a to BMSCs has been demonstrated to activate the expression of pro-tumorigenic cytokines IL-6 and CXCL1 in a gamma-secretase dependent manner (De Veirman et al., 2016).

In addition to being an active component of the MM-EV cargo, Notch2 has been demonstrated to drive expression of heparanase in malignant MM cells—an enzyme involved in regulating exosomal secretion by cleaving membrane-bound heparan sulfates (Maichl et al., 2023). Heparanase is of particular importance to MM as its primary function is to cleave the MM-defining antigen, syndecan-1 (CD138) from the plasma membrane (Akhmetzyanova et al., 2020; Fooksman, Mazumder & McCarron, 2015).

Heparanase-mediated cleavage of syndecan-1 has been implicated in exosomal biogenesis, packaging, and docking (Purushothaman & Sanderson, 2020). Mechanistically, heparanase truncates syndecan-1 intracellularly allowing binding to syntenin of the ALIX complex, this subsequently recruits the endosomal sorting complex (ESCRT) which initiates the production and parsing of intraluminal vesicles before release as exosomes (Roucourt et al., 2015).

Heparanase not only promotes the secretion of exosomes but has also been shown to regulate the contents of the exosome cargo to favor tumor progression. MM cells enriched in heparanase have been shown to contain disproportionate amounts of VEGF, HGF, syndecan-1, and heparanase in their released EVs (Thompson et al., 2013). Finally, heparanase has also been suspected to promote the assimilation of exosomes as its expression is correlated with the internalization of exosomes and target cell-exosome adhesion occurs between heparan sulfate crosslinks bridged by fibronectin (Christianson et al., 2013; Purushothaman et al., 2016).

Extracellular matrix remodeling

Alterations to the extracellular matrix (ECM) by MM cells have been demonstrated to contribute to tumor proliferation, invasiveness, and disease progression (Glavey et al., 2017). In MM, Notch signaling has been demonstrated to regulate downstream effectors to promote remodeling of the ECM in favor of tumor progression (Table 1). Interestingly, Notch has been shown to both upregulate oncogenic ECM effectors and downregulate tumor suppressive ECM effectors. ECM oncogenes of particular significance regulated by Notch are heparanase, calcyclin, avB5, and IGF. Moreover, ECM tumor suppressor genes regulated by Notch include TGFBI and C1QA. For a comprehensive view of the constellation of ECM effectors regulated by Notch and negatively correlated with patient survival see work done by Maichl et al. (2023) and Ding & Shen (2015).

Table 1 ECM-associated oncogenes and tumor suppressor genes regulated by Notch.

Gene Name	Notch	Function	Citations	
ECM Associated Oncogenes Driven by Notch	
Heparanase (HPSE)	2	Cleaves membrane proteoglycans such as syndecan-1 (CD138)	Maichl et al. (2023), Purushothaman & Sanderson (2020), Jung et al. (2016), Ramani et al. (2011)	
Regulates the assimilation of angiogenic and oncogenic factors (VEGF, IGF, HGF)	
Polarizes invasive capacity of MM cells	
Calcyclin (S100A6)	2	Calcium binding protein	Maichl et al. (2023), Glavey et al. (2017), Nedjadi et al. (2009), Leśniak & Filipek (2023)	
Associates with ANXA2 to promote invasiveness	
Activates Wnt and MAPK pathways internally	
avB5 (ITGAV/ITGBV)		Component of the VLA-5 adhesion complex	Ding & Shen (2015), Damiano et al. (1999), Kiziltepe et al. (2012)	
Promote cell adhesion-mediated drug resistance (CAM-DR) to BMSCs and vitronectin	
Insulin Growth Factor 1 (IGF1)	1	Binds extracellular IGFBP and vitronectin	Maichl et al. (2023), Murekatete et al. (2018)	
Assembles membrane complexes promoting invasiveness and proliferation	
ECM Associated Tumor Suppressors Inhibited by Notch	
Transforming Growth Factor Beta-Induced (TGFBI)	2	ECM binding protein	Maichl et al. (2023), Davies et al. (2003), Batlle & Massagué (2019), Kaiser et al. (2013), Frassanito et al. (2016)	
Regulate morphogenesis, adhesion, migration angiogenesis, and inflammation	
Represses RAS/RAF pathway in response to oncogenic stress	
Represses PI3K/AKT pathway	
Complement C1q A Chain (C1QA)	2	Involved in C1q complement complex	Maichl et al. (2023), Liang et al. (2022)	
Binds antibody/antigen complexes and initiates complement cascade	

Osteodegenerative disease

Osteolytic lesions are a hallmark of the MM disease process. Normal bone remodeling involves the activity of osteoclasts and osteoblasts which function to resorb and secrete bone, respectively. In MM, bone remodeling is dysregulated in favor of resorption, which is likely driven evolutionarily as increased osteolysis leads to the release of matrix-bound growth factors such as insulin growth factor and transforming growth factor beta-1 (IGF and TGFB-1) (Coniglio, 2018; Morris & Edwards, 2021).

Osteoclasts are multinucleated cells derived from the fusion of macrophages when activated by the growth factor RANKL (Compton & Lee, 2014). Previous research has demonstrated that osteoclast differentiation (osteoclastogenesis) is amplified by Jagged1 stimulation of the Notch signaling pathway—primarily through the Notch2 receptor post commitment with RANKL (Ashley, Ahn & Hankenson, 2015; Fukushima et al., 2008; Colombo et al., 2014). In vitro and ex vivo research have demonstrated that MM-osteoclast cocultures promote increased osteoclast viability, multinucleation, and resorption forming ability in a gamma-secretase dependent fashion (Colombo et al., 2014; Schwarzer et al., 2008). Additionally, in vivo models have demonstrated that GSI treatment reduced tumor burden and osteolytic lesions in both cortical and trabecular bone (Schwarzer et al., 2014). Mechanistically, this has been suggested to occur through increased expression of the osteoclast target genes: NFATc1, CTSK, and TRAP5. Further research has shown that in vitro Jagged1/Jagged2 knockdown in cocultured MM cells decreases pre-osteoclast expression of Notch2, TRAP, RANK, and their autocrine expression of RANKL (Colombo et al., 2014; Schwarzer et al., 2008, 2014).

Osteoblasts are bone-secreting cells derived from mesenchymal stem cells (MSCs) and complete their life cycle by transforming into matrix-embedded osteocytes (Franz-Odendaal, Hall & Witten, 2006). Previous research has demonstrated that signaling through Notch1 inhibits osteoblast differentiation by increasing expression of sclerostin—an inhibitor of the osteogenic Wnt signaling (Zanotti & Canalis, 2016). During the transition from MGUS to MM, Notch signaling becomes active in multiple myeloma mesenchymal stem cells (MM-MSCs) which consequently suppresses osteogenesis (Fig. 3) (Xu et al., 2012; Guo et al., 2017). Ex vivo research has shown that in a Notch1 and gamma-secretase mediated fashion, MM inhibits expression of the keystone osteoblast transcription factor, RUNX2 (Yavropoulou & Yovos, 2007). Notch signaling has also been shown to inhibit other markers of osteogenesis such as BSP, DLX5, OSX, ALP, OCN, OPN, and COL-1 (Xu et al., 2012; Guo et al., 2017). Further mechanistic research has demonstrated that Notch-mediated reduction in osteoblast differentiation may be attributed to miR-223 loss of function—as its absence has been demonstrated to be driven by the presence of MM cells in a Jagged2-dependent manner (Berenstein et al., 2016).

Figure 3 Notch-mediated tumor microenvironment remodeling.

Multiple myeloma cells interact with the bone marrow stroma through Notch signaling to facilitate disease progression. These interactions promote angiogenesis, and osteoclastogenesis and reduce osteoblast differentiation. The cytokine secretion of osteocytes and mesenchymal stem cells is further altered by Notch to create a favorable environment for tumor growth. Adapted from Biorender.com.

BMSCs primarily consist of fibroblasts and mesenchymal stem cells (MSCs) (Miari & Williams, 2023). In MM, Notch-mediated interactions with BMSCs have been shown to drive the expression of pro-tumorigenic and pro-osteoclastogenic growth factors. Ex vivo studies have demonstrated that knockdown of Jagged1/2 in MM cells reduces RANKL and IL-6 expression by BMSCs (Fig. 3) (Colombo et al., 2014, 2016). MM-MSCs have also been shown to be enriched in Notch ligands and receptors, namely, Notch1/2, Jagged1, and DLL1 (Xu et al., 2012; Guo et al., 2017). This may suggest that Jagged1/2 over-expressing MM cells induce the expression of Notch ligands and receptors in MM-MSCs akin to the developmental process of Notch, lateral induction. This process is characterized by ligand over-expressing cells inducing cascades of Notch activation and expression amongst neighboring cells (Boareto et al., 2015).

Autonomous interactions between malignant MM cells contribute to the perturbation of bone remodeling through Notch-mediated overexpression of RANKL and IL-6 as knockdown of Jagged1/Jagged2 in MM reduces their expression (Houde et al., 2004; Colombo et al., 2014, 2016). In other osteodegenerative disease states, IL-6 has been shown to drive BMSC expression of RANKL and pre-osteoclast expression of RANK leading to both overexpression and potentiation of the RANKL/RANK axis (Harmer, Falank & Reagan, 2019; Menaa et al., 2000; Hashizume, Hayakawa & Mihara, 2008).

Osteocytes are cells embedded within mineralized bone and act as regulators of osteoclasts and osteoblasts (Franz-Odendaal, Hall & Witten, 2006; Delgado-Calle & Bellido, 2015). In vivo and ex vivo evidence in MM has demonstrated that activation of Notch3 in osteocytes contributes to osteolytic disease by the overexpression of RANKL and the osteoblast inhibitor sclerostin. Interestingly, Notch3 activation appears to be reciprocal between malignant cells and osteocytes; as Notch3 activation in MM leads to enhanced proliferation and MM expression of RANKL (Delgado-Calle et al., 2016; Sabol et al., 2022).

Angiogenesis

Tumor vascularization, or angiogenesis, has been shown to be stimulated by Notch in MM. In ex vivo and in vivo knockdown studies, interactions between Jagged1/2 and Notch1/2 have been demonstrated to promote adhesion, migration, and organization of multiple myeloma endothelial cells (MMECs) (Palano et al., 2020; Saltarella et al., 2019). Interestingly, overexpression of Jagged1/2 was demonstrated to be a prognostic biomarker in MMECs as compared to MGUSECs (Saltarella et al., 2019). Mechanistically, both Notch interactions between MM cells on MMECs and autonomous interactions amongst MMECs lead to the expression of hepatocyte growth factor (HGF), vascular endothelial growth factor (VEGF) and the VEGF receptor (VEGFR) (Fig. 3) (Palano et al., 2020; Saltarella et al., 2019). Moreover, autonomous Notch interactions between MM cells and MM interactions with BMSCs drive the expression of pro-angiogenic factors (Fig. 3). Namely, MM Jagged1/2 knockdown studies have shown reduced MM expression of VEGF and Jagged2/Notch1 interactions have shown to drive expression of VEGF and IGF-1 in BMSCs (Houde et al., 2004; Palano et al., 2020; Berenstein et al., 2016).

Drug resistance

Mounting evidence demonstrates that downstream effectors of Notch signaling confer resistance to many small molecule therapeutics in MM (Table 2). Lenalidomide is an immunomodulatory agent that induces the degradation of the key MM transcription factors IKZF1/IKZF3 and activate adaptive immune responses against MM (Krönke et al., 2014). Notch1 and 2 have been demonstrated to confer resistance to lenalidomide through increased compensatory colony forming (Chang et al., 2023; Maichl et al., 2023; Colombo et al., 2020). Mechanistically, this has been inferred to occur through Notch-mediated upregulation of the CXCR4/SDF-1 axis which drives overexpression of drug-resisting effectors BCL2, survivin, and ABCC1 (Colombo et al., 2020; Cho et al., 2011). BCL2 and survivin resist apoptosis by inhibiting BAX/BAK mediated release of cytochrome C and caspase9 activity, respectively while ABCC1 promotes the efflux of therapeutic compounds. Upregulation of these genes has been characterized to induce cancer stem cell behavior, which further corroborates the role of Notch in tumor initiation (Waclawiczek et al., 2023; Garg et al., 2016; Reed, 2008; Begicevic & Falasca, 2017).

Table 2 Pro-proliferation, drug resistance, and tumor suppressor genes dysregulated by Notch in MM.

Gene	Resistance	Mechanism	Citations	
Pro-proliferation and Drug Resistance Effector Genes Driven by Notch	
CXCR4/SDF1a	Bor, Mel, Len	Upregulates ABCC1, survivin and Bcl2	Maichl et al. (2023), Berenstein et al. (2016), Miari & Williams, (2023), Palano et al. (2020)	
Promotes BM chemotaxis	
p21	Bor, Mel, Mit	Promotes growth arrest and survival against replication targeting drugs	Ishibashi et al. (2020), Reed (2008)	
CyclinD1/2	Bor, Mel	Drives cell cycle from G1 to S phase	Ishibashi et al. (2020)	
CYP1A1	Bor	Promotes metabolic degradation of drugs	Menaa et al. (2000), Muguruma et al. (2017)	
avB5	Dox	Assembles VLA-5 adhesion complexes	Ding & Shen (2015)	
Promotes BMSC and vitronectin binding	
MARCK	Bor	Promotes cell cycle progression during drug-induced cell cycle arrest	Saltarella et al. (2019), Berenstein et al. (2016)	
IL-6	Bor	Resists apoptosis through STAT/Bcl2 Promotes proliferation through Ras/MAPK and PI3K/AKT	Colombo et al. (2016), Abramson (2023), Bhatt, Kloock & Comenzo (2023), Cho et al. (2011)	
Note:

Bor, bortezomib; dox, doxorubicin; mel, melphalan; mit, mitoxantrone.

Bortezomib is a 20S proteasomal subunit inhibitor that confers MM toxicity by allowing immunoglobulin accumulation and the terminal protein response (Abramson, 2023; Obeng et al., 2006). Signaling through multiple Notch axes has been demonstrated to confer bortezomib resistance through CXCR4/SDF1, CYP1A1, MARCKs, IL-6, cyclin D1/2, and in a context-dependent fashion, p21 (Maichl et al., 2023; Colombo et al., 2020; Xu et al., 2012). Jagged-mediated interactions regulate most of these effectors leading to CXCR4/SDF1 axis upregulation and expression of BCL2, survivin, and ABCC1 which confer drug resistance to bortezomib similarly to the effects observed with lenalidomide (Chang et al., 2023; Colombo et al., 2020; Azab et al., 2009). Myristolated alanine-rich C kinase substrates (MARCKs)—known to be overexpressed in MM—contribute to bortezomib resistance by counteracting drug-mediated cell cycle arrest through the E2F signaling complex and inhibiting p27 (Yang et al., 2015; Muguruma et al., 2017). IL-6 not only serves as a necessary growth factor for MM cells but is also purported to confer bortezomib resistance by inhibiting caspase activity (Colombo et al., 2016; Farrell & Reagan, 2018; Liu et al., 2022). Additionally, compensatory cyclin D1/D2 expression confer bortezomib resistance by promoting G1 to S phase cell cycle progression (Ishibashi et al., 2020). BMSC-derived DLL1 interactions with MM Notch2 have been shown to confer drug resistance to bortezomib through upregulation of the drug-metabolizing enzyme cytochrome p45 (CYP1A1) (Xu et al., 2012; Percy & Percy, 2015).

Dexamethasone is a synthetic adrenocorticosteroid—25 times stronger than cortisol—which functions by binding to the glucocorticoid receptor thereby inhibiting downstream NFKB signaling and IL-6 expression (Rosenberg, 2023; Nissen & Yamamoto, 2000). Notch dysregulation due to p53 deletion has been shown to be associated with dexamethasone resistance which may confer drug resistance through downstream overexpression of CXCR4/SDF1 signaling (Chang et al., 2023; Colombo et al., 2020; Azab et al., 2009).

Doxorubicin and mitoxantrone are DNA replication inhibitors that function by inhibiting topoisomerase II (Bhatt, Kloock & Comenzo, 2023; Faulds et al., 1991). Signaling through Notch1 has been demonstrated to confer resistance to doxorubicin and mitoxantrone by regulating p21, HES, and as previously mentioned, avB5 (Chang et al., 2023; Ishibashi et al., 2020; Nefedova et al., 2004, 2008). These effectors work by providing strategic growth arrest which exploits the replication-dependent mechanics of these drugs. Regulation of these effectors may work in an orchestrated manner where avB5 complexes create CAM-DR while p21 induces strategic growth arrest (Ding & Shen, 2015; Fu et al., 2023). Finally, HES works to inhibit NOXA—a proapoptotic facilitator of BAX/BAK-mediated cell death (Ploner, Kofler & Villunger, 2008; Li et al., 2010).

Melphalan is an alkylating agent that induces cytotoxicity by creating inter and intra-cross links in DNA (Abramson, 2023; Poczta, Rogalska & Marczak, 2021). Both Notch1/2 and pan Notch signaling have been demonstrated to confer resistance to melphalan. In line with previously cited works it is suspected that these resistance mechanisms may be established by HES1, cyclin D1/2, p21, and CXCR4/SDF1 signaling with its downstream effectors ABCC1, survivin, and BCL2 (Chang et al., 2023; Ishibashi et al., 2020; Colombo et al., 2020; Nefedova et al., 2008).

Targeting Notch therapeutically

Numerous mechanisms of targeting Notch have been pursued. This review will focus on the current and promising biological concepts of targeting the pathway for MM and other malignant diseases (Table 3). For comprehensive reviews on the current state of prospective Notch targeting drugs and recent clinical trials see reviews by Medina et al. (2023) and Li et al. (2023). Promising concepts include gamma-secretase inhibitors, transcriptional complex inhibitors, monoclonal antibodies, and high affinity engineered Delta-like-ligand decoy ligands. GSIs are of particular relevance to MM as they may function synergistically with BCMA-targeting immunotherapies—as BCMA is one of many substrates cleaved by gamma-secretase. While novel GSIs appear to be promising, 149 known gamma-secretase catalyzed substrates have been documented and because of this, targeting Notch through gamma-secretase risks off-target effects (Güner & Lichtenthaler, 2020; Merilahti & Elenius, 2019; Haapasalo & Kovacs, 2011).

Table 3 Notch targeting therapeutics.

Notch Targeting Therapeutics	
Drug	Disease Context	Phase	Adverse Events	Citations	
Gamma Secretase Inhibitors	
MK-0752	Various solid tumors	Phase I clinical trial	GI toxicity	Krop et al. (2012)	
RO4929097	Melanoma	Phase II clinical trial	GI toxicity	Lee et al. (2015)	
MRK-560	T-ALL	Mouse xenograft model	Reduced GI toxicity	Habets et al. (2019)	
Selectively targets PSEN1 of gamma-secretase	
MRK003	NHL	In vitro	Increased expression of RELB oncogene	Ramakrishnan et al. (2012)	
MM	
BT-GSI	MM	Mouse xenograft model	Reduced GI toxicity	Sabol et al. (2021)	
NICD-MAML-CSL Transcriptional Complex Inhibitors	
SAHM1	T-ALL	Mouse xenograft model	Reduced GI toxicity	Moellering et al. (2009)	
MAML competitive inhibitor	
CB-103	Various cancers	Phase I/II clinical trials	Reduced GI toxicity	Medina et al. (2023), Li et al. (2023)	
Notch Ligand/Receptor Monoclonal Antibodies	
Bronctictuzumab	Various cancers	Phase I clinical trials	GI toxicity	Casulo et al. (2016), Ferrarotto et al. (2018)	
Targets Notch1	
Tarextumab	Various solid tumors	Phase I/II clinical trials	GI toxicity	Daniel et al. (2017), Smith et al. (2019)	
Targets Notch2/3	
CTX104	EG-7 lymphoma	Mouse xenograft model	Reduced GI toxicity	Sierra et al. (2017)	
Targets Jagged1/2	
15D11	Breast cancer	Mouse xenograft model	Reduced GI toxicity	Masiero et al. (2019), Zheng et al. (2017)	
Targets Jagged1	
Engineered Notch Ligands	
Soluble DLL4	High affinity	Mouse xenograft	Reduced toxicity	Gonzalez-Perez et al. (2023)	
DLL4 ligand	

Conclusions

The majority of multiple myeloma (MM) cases (92%) overexpress Notch1 and Jagged1 and expression of these genes correlate with disease progression. While the causes of this overexpression are unclear, it appears linked to genetic factors such as chromosome 17/p53 deletions and obscure disruptions of transcriptional networks.

Dysregulated Notch signaling promotes MM through several mechanisms: (1) enhanced proliferation via IL-6 and cyclin D upregulation, (2) increased tumor-initiating potential by inhibiting p21 and p27, (3) activation of pro-survival pathways like CXCR4/SD-1, (4) alterations enabling bone marrow infiltration and drug resistance, and (5) release of Notch components in exosomes. Normally, Notch1/2 expression is lost after germinal center B cell differentiation, but persistence in MM plasma cells likely drives abnormal proliferation.

Interactions between MM cells and the tumor microenvironment further exacerbate the disease. Notch signaling (1) increases osteoclastogenesis and (2) suppresses osteoblast differentiation, contributing to bone degradation. It also (3) enhances angiogenesis and (4) reprograms the cytokine environment, promoting growth factors like VEGF, Il-6, SDF1, RANKL, IGF, and sclerostin.

Remarkably, Notch signaling contributes to global changes in gene expression in malignant MM cells and cells of the tumor microenvironment; this dysregulation appears to influence many clinically relevant aspects of MM. While ostensibly a logical therapeutic target for MM, a more extensive understanding of the mechanics of Notch signal transduction, the transcriptional landscape regulated by Notch, and a more precise understanding of the etiology of Notch dysregulation in MM must be elucidated to operationalize the Notch signaling pathway for safe and personalized translational implementation.

Special thanks to Dr. Judd Case, Dr. Jason Ashley, and Dr. Justin Bastow for guidance and oversight contributed to this article.

Additional Information and Declarations

Competing Interests

Author Contributions

Data Availability

The author declares that they have no competing interests.

Justin Roosma conceived and designed the experiments, performed the experiments, analyzed the data, prepared figures and/or tables, authored or reviewed drafts of the article, and approved the final draft.

The following information was supplied regarding data availability:

This is a literature review.

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
