# Peer review of "A comprehensive review of oncogenic Notch signaling in multiple myeloma"

_PeerJ, doi:10.7717/peerj.18485_

## Round 0.1 · original submission · Minor Revisions

Dear Dr. Roosma,

Thank you for your submission to PeerJ. We have received review reports from a panel of expert reviewers and all of them suggest only minor revisions. I request you to resubmit the manuscript after addressing the reviewer's comments and highlighting the changes.

Regards,

·

Basic reporting

No Comments.

Experimental design

No Comments

Validity of the findings

No Comments

Reviewer 2 ·

Basic reporting

The entire manuscript should be modified in to readable format.

Experimental design

Study design can be acceptable

Validity of the findings

no comment

Additional comments

The concept is good and requires modification with a proper aim to study.

Annotated reviews are not available for download in order to protect the identity of reviewers who chose to remain anonymous.

·

Basic reporting

There are a number of review articles on Notch signaling in cancer and a few in regards to multiple myeloma (linked below). What are some key differences that this review brings to the field?
“The multifunctional role of Notch signaling in multiple myeloma” https://www.ncbi.nlm.nih.gov/pmc/articles/PMC8589324/
“Notch signaling deregulation in multiple myeloma: A rational molecular target” https://www.ncbi.nlm.nih.gov/pmc/articles/PMC4694956/

Experimental design

Line 32-63: Update these estimates for 2024.
Line 254-267: Provide references to the in vitro studies

Validity of the findings

No comment

Additional comments

The author has done a thorough review of the literature on Notch signaling in multiple myeloma. It is very well written and well organized.

Reviewer 4 ·

Basic reporting

The article is well-written and appropriately covers the topic of Notch signaling and its potential role in multiple myeloma development, pathogenesis, and treatment. The article is less successful summarizing the current state of myeloma and uses some antiquated and obsolete definitions in its review of the current state of myeloma. For example, the definition of plasma cell leukemia was changed to 5% or more circulating plasma cells in 2021 (see Fernandez de Larrea et al Blood Cancer Journal 2021). Similarly, the diagnostic criteria for myeloma do not mention the SLiM criteria which was added to the definition of myeloma in 2014. Additionally, the epidemiologic statistics cited are outdated and newer (and more relevant) data are published yearly. As this is a review article (which should summarize all the data available at the time of submission), I would highly encourage the authors to revise the manuscript with newer data.

Experimental design

As this is a review article the methods are appropriate. As mentioned previously some of the epidemiologic, diagnostic, and prognostic discussions of myeloma are antiquated and I would recommend updating the data, diagnostic criteria, and references accordingly.

Validity of the findings

The conclusion section is overly long and is more of a summary of the data presented than a true conclusion. Still, the authors make a compelling argument for further study of the notch pathway as it pertains to myeloma.

Additional comments

Figure 1 and Figure 2 should have their names switched as figure 2 is referenced in the text prior to figure 1.

Reviewer 5 ·

Basic reporting

This is a nice review of the importance of Notch in MM.

My only comment is in Line 120-125 the important 2ndry mutations should mention activating myc events, which are common and important in MM.

Experimental design

No comment, review article

Validity of the findings

Well written and brings out the important points

Additional comments

NIl

---

## Round 0.2 · accepted · Accept

All issues pointed by the reviewers were adequately addressed, and revised manuscript is acceptable now.

Reviewer 2 ·

Basic reporting

no comment

Experimental design

Finds good and considerable

Validity of the findings

no comment

Additional comments

All the queries raised by me were answered appropriately

·

Basic reporting

no comment

Experimental design

no comment

Validity of the findings

no comment

Reviewer 5 ·

Basic reporting

No additional comments

Experimental design

No additional comments

Validity of the findings

No additional comments

Additional comments

No additional comments